# To ArXiv or not to ArXiv: A Study Quantifying Pros and Cons of Posting Preprints Online

## Abstract

Double-blind conferences have engaged in debates over whether to allow authors to post their papers online on arXiv or elsewhere during the review process. Independently, some authors of research papers face the dilemma of whether to put their papers on arXiv due to its pros and cons. We conduct a study to substantiate this debate and dilemma via quantitative measurements. Specifically, we conducted surveys of reviewers in two top-tier double-blind computer science conferences—ICML 2021 (5361 submissions and 4699 reviewers) and EC 2021 (498 submissions and 190 reviewers). Our three main findings are as follows. First, more than a third of the reviewers self-report searching online for a paper they are assigned to review. Second, conference policies restricting authors from publicising their work on social media or posting preprints before the review process may have only limited effectiveness in maintaining anonymity. Third, outside the review process, we find that preprints from better-ranked institutions experience a very small increase in visibility compared to preprints from other institutions.

## 1 Introduction

Across academic disciplines, peer review is used to decide on the outcome of manuscripts submitted for publication. Single-blind reviewing used to be the predominant method in peer review, where the authors' identities are revealed to the reviewer in the submitted paper. However, several studies have found various biases in single-blind reviewing. These include bias pertaining to affiliations or fame of authors (Blank, 1991; Sun et al., 2021; Manzoor & Shah, 2021), bias pertaining to gender of authors (Rossiter, 1993; Budden et al., 2008; Knobloch-Westerwick et al., 2013; Roberts & Verhoef, 2016), and others (Link, 1998; Snodgrass, 2006; Tomkins et al., 2017). These works span several fields of science and study the effect of revealing the authors' identities to the reviewer through both observational studies and randomized control trials. These biases are further exacerbated due to the widespread prevalence of the Matthew effect—rich get richer and poor get poorer—in academia (Merton, 1968; Squazzoni & Claudio, 2012; Thorngate & Chowdhury, 2013). Biases based on authors' identities in peer review, coupled with the Matthew effect, can have far-reaching consequences on researchers' career trajectories.

As a result, many peer-review processes have moved to *double-blind* reviewing, where authors' names and other identifiers are removed from the submitted papers. Ideally, in a double-blind review process, neither the authors nor the reviewers of any papers are aware of each others' identity. However, a challenge for ensuring that reviews are truly double-blind is the exponential growth in the trend of posting papers online before review (Xie et al., 2021). Increasingly, authors post their preprints on online publishing websites such as arXiv and SSRN and publicize their work on social media platforms. The conventional publication route via peer review is infamously long and time-consuming, and online preprint-publishing venues provide a platform for sharing research with the community usually without delays. Not only does this help science move ahead faster, but it also helps researchers avoid being "scooped". However, the increase in popularity of making papers publicly available—with author identities—before or during the review process, has led to the dilution of double-blinding in peer review. For instance, the American Economic Association, the flagship journal in economics, dropped double-blinding in their reviewing process citing its limited effectiveness in maintaining anonymity. The availability of preprints online presents a challenge in double-blind reviewing,

which could lead to biased evaluations for papers based on their authors' identities, similar to single-blind reviewing.

This dilution has led several double-blind peer-review venues to debate whether authors should be allowed to post their submissions on the Internet, before or during the review process. For instance, top-tier machine learning conferences such as NeurIPS and ICML do not prohibit posting online. On the other hand, the Association of Computational Linguistics (ACL) had introduced a policy in 2018 (repealed in January 2024) for its conferences in which authors were prohibited from posting their papers on the Internet starting a month before the paper submission deadline till the end of the review process. The Conference on Computer Vision and Pattern Recognition (CVPR) has banned the advertisement of submissions on social media platforms for the same time period, since 2021. Some venues are stricter, for example, the IEEE Communication Letters and IEEE International Conference on Computer Communications (INFOCOMM) disallows posting preprints to online publishing venues before acceptance.

Independently, authors who perceive they may be at a disadvantage in the review process if their identity is revealed face a dilemma regarding posting their work online. On one hand, if they post preprints online, they are at risk of de-anonymization in the review process, if their reviewer searches for their paper online. Past research suggests that if such de-anonymization happens, reviewers may get biased by the author identity, with the bias being especially harmful for authors from less prestigious organizations. On the other hand, if they choose to not post their papers online before the review process, they stand to lose out on viewership and publicity for their papers.

It is thus important to quantify the consequences of posting preprints online to (i) enable an evidence-based debate over conference policies, and (ii) help authors make informed decisions about posting preprints online. In our work, we conduct a large-scale survey-based study in conjunction with the review process of two top-tier publication venues in computer science that have double-blind reviewing: the 2021 International Conference on Machine Learning (ICML 2021) and the 2021 ACM Conference on Economics and Computation (EC 2021).[1] Specifically, we design and conduct experiments aimed at answering the following research questions:

(Q1) What fraction of reviewers, who had not seen the paper they were reviewing before the review process, deliberately search for the paper on the Internet during the review process?

(Q2) What are the trends in posting preprints online and the visibility enjoyed by these preprints? Consequently, what are their implications for conference policies concerning posting submissions online?

(Q3) Given a preprint is posted online, what is the causal effect of the rank of the authors' affiliations on the visibility of a preprint to its target audience?

Our work can help inform authors' choices of posting preprints as well as enable evidence-based debates and decisions on conference policies. Our results also inform conference policies with data-driven insights, that were previously primarily driven by opinions and anecdotal evidence.

Finally, we list the main findings and recommendations from our research study and analysis:

- **[Q1] Searching for paper online**
    - More than a third of survey respondents self-report searching for their assigned papers online.
    - In double-blind review processes, reviewers should be explicitly instructed to not search for their assigned papers on the Internet.

- **[Q2] Trends and visibility of preprints**
    - On average, authors posting preprints online receive viewership from 8% of relevant researchers in ICML and 20% in EC before the conclusion of the review process.

---

[1]In Computer Science, conferences are typically the terminal publication venue and are typically ranked at par or higher than journals. Full papers are reviewed in computer science conferences, and their publication has archival value.

– Certain conferences ban posting preprints a month before the submission deadline. In ICML and EC (which did not have such a ban), more than 50% of preprints posted online where posted before the one month period, and these enjoyed a visibility of 8.6% and 18% respectively.

– Conference policies designed towards banning authors from publicising their work on social media or posting preprints before the review process may have only limited effectiveness in maintaining anonymity.

- **[Q3] Authors' affiliations and visiblity**

  – For posted preprints online, authors from lower-ranked institutions experience only a very small reduction in visibility compared to authors from top-ranked institutions, 0.02 in ICML and 0.04 in EC when measuring visibility on a scale of 0 to 1. This suggests that the benefits of posting preprints online are nearly comparable for both groups.

## 2   Related work

*Surveys of reviewers.*  Several studies survey reviewers to obtain insights into reviewer perceptions and practices. Nobarany et al. (2016) surveyed reviewers in the field of human-computer interaction to gain a better understanding of their motivations for reviewing. They found that encouraging high-quality research, giving back to the research community, and finding out about new research were the top general motivations for reviewing.  Along similar lines, Tite & Schroter (2007) surveyed reviewers in biomedical journals to understand why peer reviewers decline to review. Among the respondents, they found the most important factor to be conflict with other workload.

 Resnik et al. (2008) conducted an anonymous survey of researchers at a government research institution concerning their perceptions about ethical problems with journal peer review. They found that the most common ethical problem experienced by the respondents was incompetent review. Additionally, 6.8% respondents mentioned that a reviewer breached the confidentiality of their article without permission. This survey focused on the respondents' perception, and not on the actual frequency of breach of confidentiality. In another survey, by Martinson et al. (2005), 4.7% authors self-reported publishing the same data or results in more than one publication. Fanelli (2009) provides a systematic review and meta analysis of surveys on scientific misconduct including falsification and fabrication of data and other questionable research practices.

Goues et al. (2018) surveyed reviewers in three double-blind conferences to investigate the effectiveness of anonymization of submitted papers. In their experiment, reviewers were asked to guess the authors of the papers assigned to them. Out of all reviews, 70%-86% of the reviews did not have any author guess. Here, absence of a guess could imply that the reviewer did not have a guess or they did not wish to answer the question. Among the reviews containing guesses, 72%-85% guessed at least one author correctly.

*Analyzing papers posted versus not posted on arXiv.*  Bharadhwaj et al. (2020) aim to analyse the risk of selective de-anonymization through an observational study based on open review data from the International Conference on Learning Representations (ICLR). The analysis quantifies the risk of de-anonymization by computing the correlation between papers' acceptance rates and their authors' reputations separately for papers posted and not posted online during the review process. This approach however is hindered by the confounder that the outcomes of the analysis may not necessarily be due to de-anonymization of papers posted on arXiv, but could be a result of higher quality papers being selectively posted on arXiv by famous authors. Moreover, it is not clear how the paper draws conclusions based on the analysis presented therein. Our supporting analysis overlaps with the investigation of Bharadhwaj et al. (2020): we also investigate the correlation between papers' acceptance rates and their authors' associated ranking in order to support our main analysis and to account for confounding by selective posting by higher-ranked authors.

Aman (2014) investigate possible benefits of publishing preprints on arXiv in *Quantitative Biology*, wherein they measure and compare the citations received by papers posted on arXiv and those received by papers not posted on arXiv. A similar confounder arises here that a positive result could be a false alarm due to higher quality papers being selectively posted on arXiv by authors. Along similar lines,  Feldman et al. (2018) investigate the benefits of publishing preprints on arXiv selectively for papers that were accepted for

publication in top-tier CS conferences. They find that one year after acceptance, papers that were published on arXiv before the review process have 65% more citations than papers posted on arXiv after acceptance. The true paper quality is a confounder in this analysis as well.

In our work, we quantify the risk of de-anonymization by directly studying reviewer behaviour regarding searching online for their assigned papers. We quantify the effects of publishing preprints online by measuring their visibility using a survey-based experiment querying reviewers whether they had seen a paper before.

*Studies on peer review in computer science.* Our study is conducted in two top-tier computer science conferences and contributes to a growing list of studies on peer review in computer science. Lawrence & Cortes (2014); Beygelzimer et al. (2021) quantify the (in)consistencies of acceptance decisions on papers. Several papers (Madden & DeWitt, 2006; Tung, 2006; Tomkins et al., 2017; Manzoor & Shah, 2021) study biases due to single-blind reviewing. Shah et al. (2018) study several aspects of the NeurIPS 2016 peer-review process. Stelmakh et al. (2021c) study biases arising if reviewers know that a paper was previously rejected. Stelmakh et al. (2021b) study a pipeline for getting new reviewers into the review pool. Stelmakh et al. (2020) study herding in discussions. Stelmakh et al. (2022) study citation bias in peer review. A number of recent works (Charlin & Zemel, 2013; Stelmakh et al., 2021a; Kobren et al., 2019; Jecmen et al., 2020; Noothigattu et al., 2021) have designed algorithms that are used in the peer-review process of various computer science conferences. See Shah (2021) for an overview of such studies and computational tools to improve peer review.

## 3 Experiment design

We now outline the design of the experiment that we conducted to investigate the research questions in this work. In this section, we introduce the two computer science conferences ICML 2021 and EC 2021 that formed the venues for our investigation, and provide details regarding the research questions posed in this work in the context of these two conferences and the experiments conducted therein.

**Experiment setting.** The study was conducted in the peer-review process of two conferences:

- **ICML 2021** International Conference on Machine Learning is a flagship machine learning conference. ICML is a large conference with 5361 submissions and 4699 reviewers in its 2021 edition.

- **EC 2021** ACM Conference on Economics and Computation is the top conference at the intersection of Computer Science and Economics. EC is a relatively smaller conference with 498 submissions and 190 reviewers in its 2021 edition.

Importantly, the peer-review process in both conferences, ICML and EC, is organized in a double-blind manner, defined as follows. In a **double-blind peer-review process**, the identity of all the authors is removed from the submitted papers. No part of the authors' identity, including their names, affiliations, and seniority, is available to the reviewers through the review process. At the same time, no part of the reviewers' identity is made available to the authors through the review process.

We now formally define some terminology used in the research questions: Q1, Q2, and Q3, and following each definition we provide the procedure used to capture it in data in our experiments.

### 3.1 Definitions and experiment design for Q1

The research question Q1 focuses on the fraction of reviewers who deliberately search for their assigned paper on the Internet during the conference review process. To measure this number we surveyed the reviewers in ICML 2021, and separately the reviewers in EC 2021. Importantly, as reviewers may not be comfortable answering questions about deliberately breaking the double-blindness of the review process, we designed the survey to be anonymous. At the same time, it is useful to note that neither of these two conferences explicitly had any guidelines for reviewers on (not) searching for their assigned paper online. We used the Condorcet Internet Voting Service (CIVS) (Myers, 2003), a widely used service to conduct secure and anonymous surveys. Further, we took some steps to prevent our survey from spurious responses (e.g., multiple responses

from the same reviewer). For this, in EC, we generated a unique link for each reviewer that accepted only one response. In ICML we generated a link that allowed only one response per IP address and shared it with reviewers asking them to avoid sharing this link with anyone.[2] The survey form was sent out to the reviewers via CIVS after the initial reviews were submitted. In the e-mail, the reviewers were invited to participate in a one-question survey on the consequences of publishing preprints online. The survey form contained the following question:

"During the review process, did you search for any of your assigned papers on the Internet?"

with two possible options: *Yes* and *No*. The respondents had to choose exactly one of the two options. To ensure that the survey focused on reviewers deliberately searching for their assigned papers, right after the question text, we provided additional text: "Accidental discovery of a paper on the Internet (e.g., through searching for related works) does not count as a positive case for this question. Answer *Yes* only if you tried to find an assigned paper itself on the Internet."

Following the conclusion of the survey, CIVS combined the individual responses, while maintaining anonymity, and provided the total number of *Yes* and *No* responses received. The average number of *Yes* responses in each conferences was used as the estimate of the value sought in research question Q1.

### 3.2 Definitions and experiment design for Q2 and Q3

Research questions Q2 and Q3 both focus on the visibility of a paper available on the Internet before the review process, and trends therein. In what follows, we explicitly define the terms used in Q2 and Q3 in the context of our experiments—preprint, rank associated with a paper, target audience of a paper, and visibility of a preprint—and our approach to measuring them.

**Preprint.** To study the visibility of papers released on the Internet before publication, we checked whether each of the papers submitted to the conference was available online. Specifically, for EC, we manually searched for all submitted papers to establish their presence online. On the other hand, for ICML, owing to its large size, we checked whether a submitted paper was available on arXiv (`arxiv.org`). ArXiv is the predominant platform for pre-prints in machine learning; hence we used availability on arXiv as a proxy indicator of a paper's availability on the Internet.

**Rank associated with a paper.** In this paper, the rank of an author's affiliation is a measure of author's prestige that, in turn, is transferred to the author's paper. We determine the rank of affiliations in ICML and EC based on prominently available rankings of institutions in the respective research communities. Specifically, in ICML, we rank (with ties) each institution based on the number of papers published in the ICML conference in the preceding year (2020) with at least one author from that institution (Ivanov, 2020). On the other hand, since EC is at the intersection of two fields, economics and computation, we merge three rankings—the QS ranking for computer science (QS, 2021a), the QS ranking for economics and econometrics (QS, 2021b), and the CS ranking for economics and computation (CSRankings, 2021)—by taking the best available rank for each institution to get our ranking of institutions submitting to EC. By convention, better ranks, representing more renowned institutions, are represented by lower numbers; the top-ranked institution for each conference has rank 1. Finally, we define the rank of a paper as the rank of the best-ranked affiliation among the authors of that paper. Due to ties in rankings, we have 37 unique rank values across all the papers in ICML 2021, and 66 unique rank values across all the papers in EC 2021.

**Paper's target audience.** For any paper, we define its target audience as members of the research community that share similar research interests as that of the paper. In each conference, a 'similarity score' is computed between each paper-reviewer pair, which is then used to assign papers to reviewers. We used the same similarity score to determine the target audience of a paper (among the set of reviewers in the conference). We describe the exact process for target audience selection in EC and ICML.

---

[2]The difference in procedures between EC and ICML is due to a change in the CIVS policy that was implemented between the two surveys.

In EC, the number of papers posted online before the end of the review process was small. To increase the total number of paper-reviewer pairs where the paper was posted online and the reviewer shared similar research interests with the paper, we created a new paper-reviewer assignment. For the new paper-reviewer assignment, for each paper we considered at most 8 members of the reviewing committee that satisfied the following constraints as its target audience—(1) they submitted a positive bid for the paper indicating shared interest, (2) they are not reviewing the given paper.

In ICML, a large number of papers were posted online before the end of the review process. So, we did not create a separate paper-reviewer assignment for surveying reviewers. Instead, in ICML, we consider a paper's reviewers as its target audience and queried the reviewers about having seen it, directly through the reviewer response form.

**Paper's visibility.** We define the visibility of a paper to a member of its target audience as a binary variable which is 1 if that person has seen this paper outside of reviewing contexts, and 0 otherwise. Visibility, as defined here, includes reviewers becoming aware of a paper through preprint servers or other platforms such as social media, research seminars and workshops. On the other hand, visibility does *not* include reviewers finding a paper during the review process (e.g., visibility does not include a reviewer discovering an assigned paper by deliberate search or accidentally while searching for references).

We designed a survey-based experiment to measure visibility as follows. For each paper, we identified some reviewers as its target audience. Using a survey with multiple choice questions, we asked the identified reviewers if they had seen the papers before outside of reviewing contexts. We describe the remaining details of the survey separately for EC 2021 and ICML 2021 in the rest of the section, due to differences in their implementation.

In EC 2021, reviewers were queried about a set of papers that they were not assigned to review, using a separate optional survey form that was emailed to them by the program chairs after the rebuttal phase and before the announcement of paper decisions. Each reviewer was shown the title of five papers and asked to answer the following question for each paper:

<blockquote>"Have you come across this paper earlier, outside of reviewing contexts?"</blockquote>

In the survey form, we provided examples of reviewing contexts as "reviewing the paper in any venue, or seeing it in the bidding phase, or finding it during a literature search regarding another paper you were reviewing." The question had multiple choices enumerated next and the reviewer could select more than one choice.

(a) I have NOT seen this paper before / I have only seen the paper in reviewing contexts

(b) I saw it on a preprint server like arXiv or SSRN

(c) I saw a talk/poster announcement or attended a talk/poster on it

(d) I saw it on social media (e.g., Twitter)

(e) I have seen it previously outside of reviewing contexts (but somewhere else or don't remember where)

(f) I'm not sure

If respondents selected one or more options from (b), (c), (d) and (e), we set the visibility to 1, and if they selected option (a), we set the visibility to 0. We did not use the response in our analysis, if the reviewer did not respond or only chose option (f).

In ICML 2021, we added a two-part question in the reviewer response form corresponding to the research question Q2. Each reviewer was required to answer either *Yes* or *No* to the following question for the paper they were reviewing:

<blockquote>"Do you believe you know the identities of the paper authors? If yes, please tell us how."</blockquote>

| | | EC 2021 | ICML 2021 |
|---|---|---|---|
| 1 | # REVIEWERS | 190 | 4699 |
| 2 | # SURVEY RESPONDENTS | 97 | 753 |
| 3 | # SURVEY RESPONDENTS WHO SAID THEY SEARCHED FOR THEIR ASSIGNED PAPER ONLINE | 41 | 269 |
| 4 | % SURVEY RESPONDENTS WHO SAID THEY SEARCHED FOR THEIR ASSIGNED PAPER ONLINE | 42% | 36% |

Table 1: Outcome of survey for research question Q1.

For the second part of the question a reviewer could select more than one choice and the choices are listed here:

(a) I was aware of this work before I was assigned to review it.

(b) I discovered the authors unintentionally while searching web for related work during reviewing of this paper

(c) I guessed rather than discovered whose submission it is because I am very familiar with ongoing work in this area

(d) I first became aware of this work from a seminar announcement, Archiv announcement or another institutional source

(e) I first became aware of this work from a social media or press posting by the authors

(f) I first became aware of this work from a social media or press posting by other researchers or groups (e.g. a ML blog or twitter stream)

If they responded *Yes* to the first part, and selected one or more options among (a), (d), (e) and (f) for the second part, then we set the visibility to 1, otherwise to 0.

## 4 Analysis and results

We now describe the analysis of the data collected to address the research question. For each research question, the analysis is followed by the results obtained. Importantly, our analysis is the same for the data collected from ICML 2021 and EC 2021 unless mentioned otherwise.

### 4.1 Q1: Fraction of reviewers deliberately searching for their assigned paper

For Q1, we directly report the numbers obtained from CIVS regarding the fraction of reviewers who searched for their assigned papers online in the respective conference. Table 1 provides the results of the survey for research question Q1. The percentage of reviewers that responded to the anonymous survey for Q1 is 16% (753 out of 4699) in ICML and 51% (97 out of 190) in EC. While the coverage of the pool of reviewers is small in ICML (16%), the number of responses obtained is large (753). As shown in Table 1, the main observation is that, in both conferences, at least a third of the Q1 survey respondents self-report deliberately searching for their assigned paper on the Internet. There is substantial difference between ICML and EC in terms of the response rate as well as the fraction of *Yes* responses received, however, the current data cannot provide explanations for these differences.

### 4.2 Q2: Statistics associated with preprints posted online

Research question Q2 concerns the trends of posting preprints online and the visibility enjoyed by these preprints. To measure the visibility enjoyed by a preprint, we take for each preprint the average over all responses received in the visibility survey described in Section 3.2. This survey had a response rate of 100% in ICML, while EC had a response rate of 55.8%.

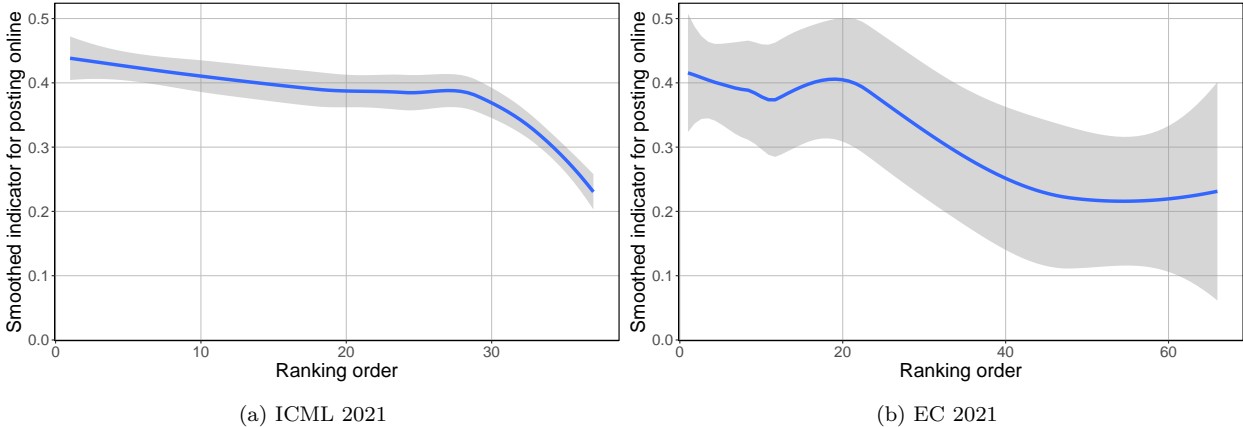

(a) ICML 2021          (b) EC 2021

Figure 1: For papers submitted to the respective conferences, we plot the indicator for paper being posted online before the end of the review process against papers' associated rank, with smoothing. On the x-axis, we order papers by their ranks (i.e., paper with the best rank gets order 1, paper with the second best rank gets order 2, and so on). The range of x-axis is given by the number of unique ranks in the visibility analysis, which may be smaller than the total number of unique ranks associated with the papers in the respective conferences. The x-axis range is 37 in Figure 1a and 40 in Figure 1b due to ties in rankings used. On the y-axis, smoothed indicator for posting online lies in $[0, 1]$. We use locally estimated smoothing to get the smoothed indicator for posting online across ranks, shown by the solid line, and a 95% confidence interval, shown by the grey region. We use local linear regression for smoothing (Cleveland & Loader, 1996)

.

Under Q2, we first discuss the general trends observed in posting preprints online, followed by observations concerning different preprint policies designed for maintaining double-anonymity in peer review.

### 4.2.1    Trends in posting preprints online and visibility

To supplement the survey data, we gathered information about when the preprints in the conferences were posted online by scraping the Internet both programmatically and manually as needed. In Table 2, we see that there were a total of 5361 and 498 papers submitted to ICML 2021 and EC 2021 respectively, out of which 1934 and 183 were posted online before the end of the review process respectively. Thus, we see that about 37% of the papers submitted were available online.

We investigate whether the pool of papers posted online before the review process is significantly different, in terms of their rank profile. Specifically, we compute the correlation (Kendall's Tau-b) between a binary value indicating whether a submitted paper was posted online before the start of the review process, and the paper's associated rank. We flip the sign of the statistic with respect to the rank variable. There is a positive correlation between a paper's associated rank and whether it was posted online before the review process in both ICML and EC of 0.12 ($p < 10^{-5}$) and 0.09 ($p = 0.01$) respectively. This suggests that the authors from higher-ranked institutions are more likely to post their papers online before the review process. In Figure 1 we provide visualization to interpret the correlation between ranking and uploading behaviour.

Next, based on results from the visibility survey, we learn that the mean visibility in ICML 2021 is 8.36% and that in EC 2021 is 20.5% (refer Table 2, Row 3). This provides an estimate of the fraction of relevant researchers in the community viewing preprints available online. Further we note that the mean visibility in ICML is considerably smaller than that in EC. This may be attributed to the following reason: The research community in EC is smaller and more tight-knit, meaning that there is higher overlap in research interests within the members of the community (reviewers). On the other hand, ICML is a large publication venue with a more diverse and spread-out research community.

| | | EC 2021 | ICML 2021 |
|---|---|---|---|
| 1 | # Papers submitted | 498 | 5361 |
| 2 | # Papers posted online before conclusion of the review process | 183 (36.7%) | 1934 (36.6%) |
| 3 | Mean visibility of preprints posted before review process conclusion | 0.21 (92/449) | 0.084 (635/7594) |
| 4 | # Papers posted online during the one month embargo period | 74 | 918 |
| 5 | Mean visibility of preprints posted during the one month embargo period | 0.24 (45/189) | 0.081 (292/3600) |
| 6 | # Papers posted online prior to the one month embargo period | 109 | 1016 |
| 7 | Mean visibility of preprints posted prior to the one month embargo period | 0.18 (47/260) | 0.086 (343/3994) |
| 8 | $p$-value for significance of difference in row 5 and row 7 | 0.15 | 0.45 |

Table 2: Rows 1-3 show general statistics regarding preprint uploading behaviour and visibility. Rows 4-8 shows these statistics pertaining to the one month embargo policy prohibiting authors from posting preprints online less than a month before the paper submission deadline till the conclusion of the review process. In the last row, the two-tailed $p-$value is obtained using Fisher's exact test.

### 4.2.2 One month embargo policy

To maintain anonymity of submitted papers, many conferences ban authors from posting preprints online for some time period before the conclusion of the review process. Notably, in the past, many conferences in the field of natural language processing such as NAACL, EMNLP, etc, had a one month embargo period which prohibited authors from posting their papers on the Internet starting a month before the paper submission deadline till the end of the review process.

To understand the potential effectiveness of such a policy in maintaining anonymity, we use the data collected from the visibility survey in ICML 2021 and EC 2021. The quantitative insights observed in the data are provided in Table 2. We see that in both ICML and EC there is no significant difference in the mean visbility of preprints posted online during the one month embargo period compared to those posted prior to that period.

### 4.2.3 Social media ban policy

Another policy designed to maintain double anonymity, seen in the recent editions of the Conference on Computer Vision and Pattern Recognition (CVPR), bans the publicity of preprints on social media websites. Recall that the visibility survey described in Section 3.2 captures how reviewers found out about the papers that we identified them to belong to the target audience of. We now analyse the responses obtained for this question to understand the potential usefulness of the social media ban policy.

Survey respondents could pick one or more from a list of options which included the option that they found a paper on social media. In Table 3 and Table 4 we tally the survey responses in EC and ICML respectively. The numbers obtained suggest that a majority of the reviewers learned about the papers from preprint servers such as arXiv. On the other hand, only a small proportion of responses mentioned social media as the source of their information about the paper. This suggests that social media might only be a second order contributor to preprint visibility.

### 4.3 Q3: Effect of the rank of the authors' affiliations on preprint visibility

We describe our analysis procedure for Q3 on investigating the causal effect of papers' ranking on their visibility. Recall that for Q3, we collected survey responses and data about papers submitted to ICML and EC that were posted online before the corresponding review process. Since the data is observational, and Q3 aims to identify the causal effect, we setup the problem and describe the causal model assumed in our setting in Section 4.3.1 followed by the corresponding analysis procedure in Section 4.3.2 and results in Section 4.3.3.

### 4.3.1 Q3 problem setup

In this section, we set up the causal estimation problem for Q3 and describe the modelling assumptions we make. To address Q3, we consider all the papers submitted to the respective conferences that were

| List of choices for question in visibility survey | Count |
|---|---|
| (a) I have NOT seen this paper before / I have only seen the paper in reviewing contexts | 359 |
| (b) I saw it on a preprint server like arXiv or SSRN | 51 |
| (c) I saw a talk/poster announcement or attended a talk/poster on it | 22 |
| (d) I saw it on social media (e.g., Twitter) | 4 |
| (e) I have seen it previously outside of reviewing contexts (but somewhere else or don't remember where) | 29 |
| (f) I'm not sure | 24 |

Table 3: Set of choices provided to reviewers in EC in visibility survey and the number of times each choice was selected in the responses obtained. There were 449 responses in total, out of which 92 responses indicated a visibility of 1.

| List of choices for question in visibility survey | Count |
|---|---|
| (a) I was aware of this work before I was assigned to review it. | 373 |
| (b) I discovered the authors unintentionally while searching web for related work during reviewing of this paper | 47 |
| (c) I guessed rather than discovered whose submission it is because I am very familiar with ongoing work in this area. | 28 |
| (d) I first became aware of this work from a seminar announcement, Archiv announcement or another institutional source | 259 |
| (e) I first became aware of this work from a social media or press posting by the authors | 61 |
| (f) I first became aware of this work from a social media or press posting by other researchers or groups (e.g. a ML blog or twitter stream) | 52 |

Table 4: Set of choices provided to reviewers in ICML in visibility survey question and the number of times each choice was selected in the set of responses considered that self-reported knowing the identities of the paper authors outside of reviewing contexts. There were a total of 635 such responses that indicated a visibility of 1. Recall that for ICML, we consider the set of responses obtained for submissions that were available as preprints on arXiv. There were 1934 such submissions.

posted online before the review process. In other words, the effect estimated would give the effect of papers' rank on visibility under the current preprint-posting habits of authors. We formally define the problem setup as follows. For any preprint $i \in \mathbb{N}^+$, we have the rank of the authors' associated affiliation, denoted by $r_i \in \mathbb{N}^+$, giving the independent variable. Each preprint after being posted online would enjoy some viewership from its target audience, this is the dependent variable in our analysis. We measure visibility, denoted by $v_i \in [0, 1]$, as a fraction between 0 and 1 where 0 implies no one in the target audience viewed the paper. To check if there is advantage provided by the associated institutional ranking on preprint viewership, we setup a hypothetical causal experiment as follows.

First we section the rank associated with a paper into two groups: low and high, such that ranks in group 'low' indicated by $r = 0$, and those in group 'high' indicated by $r = 1$ (Tomkins et al., 2017). For ICML, we draw on Tomkins et al. (2017) (which pertains to a conference in a similar field) to set the ranks in the low rank group as those under or equal to 50, and high rank group as those above 50. Meanwhile, for EC, which is a smaller conference in a different subfield, to maintain balance between the two groups, we set the rank threshold as 10. This yields a number of paper-reviewer pairs in the two groups, where a paper-reviewer pair is counted if the reviewer has provided a visibility response for the paper. In ICML, the low rank group and the high rank group contain 4936 and 2658 pairs respectively. In EC the low rank group and the high rank group contain 269 and 180 pairs respectively.

Next, we define the target estimand for the group $r = 0$ as $\mathbb{E}[\mathbf{V}^0]$ which is the expected value of the visibility enjoyed by a paper if it had a rank sampled uniformly at random from the set of 'low' ranks in our data, all else remaining the same. Similarly, we have $\mathbb{E}[\mathbf{V}^1]$ giving the expected visibility of a paper if it had a rank

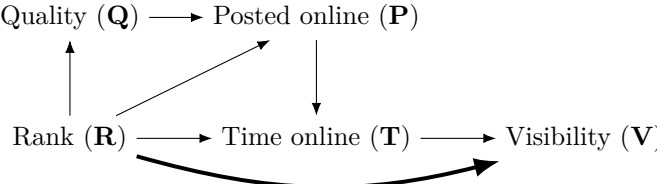

Figure 2: Graphical causal model illustrating the model assumed in our setting, to analyse the direct effect of "Rank" associated with a paper on the "Visibility" enjoyed by the paper as a preprint available on the internet.

sampled uniformly at random from the set of 'high' ranks. Finally, we measure the average treatment effect (ATE) of binarized rank on the visibility of a preprint, denoted by $\tau$, and formalized as

$$\tau = \mathbb{E}\left[\mathbf{V}^1\right] - \mathbb{E}\left[\mathbf{V}^0\right]. \tag{1}$$

Here, $\tau$ measures the expected difference in visibility for a paper under two different rank settings.

To identify the effect, Figure 2 shows the graphical causal model assumed in our setting. The model contains a paper's associated rank, the visibility enjoyed by the paper online, and three intermediate factors: (1) whether the preprint was posted online, denoted by $\mathbf{P}$; (2) the amount of time for which the preprint has been available online, denoted by $\mathbf{T}$; and (3) the objective quality of the paper, denoted by $\mathbf{Q}$. We now provide an explanation for this causal model.

First, the model captures mediation of effect of $\mathbf{R}$ on $\mathbf{V}$ by the amount of time for which the preprint has been available online, denoted by $\mathbf{T}$. For a paper posted online, the amount of time for which it has been available on the Internet can affect the visibility of the paper. For instance, papers posted online well before the deadline may have higher visibility as compared to papers posted near the deadline. Moreover, the time of posting a paper online could vary across institutions ranked differently. Thus, amount of time elapsed since posting can be a mediating factor causing indirect effect from $\mathbf{R}$ to $\mathbf{V}$. This is represented in Figure 2 by the causal pathway between $\mathbf{R}$ and $\mathbf{V}$ via $\mathbf{T}$.

Second, in the causal model, we consider the papers not posted online before the review process. For a preprint not posted online, we have $\mathbf{T} = 0$, and we do not observe its visibility. However, it is of interest that the choice of posting online before or after the review process could vary depending on both the quality of the paper as well as the rank of the authors' affiliations. For instance, authors from lower-ranked institutions may refrain from posting preprints online due to the risk of de-anonymization in the review process that could negatively influence their reviews. Or, they may selectively post their high quality papers online. This is captured in our model by introducing the variable $\mathbf{P}$. In Figure 2, this interaction is captured by the direct pathway from $\mathbf{R}$ to $\mathbf{P}$ and the pathway from $\mathbf{R}$ to $\mathbf{P}$ via $\mathbf{Q}$.

Finally, we explain the missing edges in our causal model. In the model, we assume that there is no causal link between $\mathbf{Q}$ and $\mathbf{V}$, this assumes that the initial viewership achieved by a paper does not get affected by its quality. This assumption follows from the observations made in Section 4.2.3. The responses tallied in Table 3 and Table 4 indicate that a majority of the reviewers learnt about the preprints they were queried about from first hand sources such as arXiv and talk announcements. This finding suggests that on most occasions people view a paper (and its authors) before knowing the paper's quality. Thus, in our model, we assume that the role of the quality of the paper in its visibility is absent. Next, we assume that there is no link from $\mathbf{P}$ to $\mathbf{V}$ since the variable $\mathbf{T}$ captures all the information from $\mathbf{P}$. There is no causal link from $\mathbf{Q}$ to $\mathbf{T}$. Here, we assume that $\mathbf{P}$ captures all the information of $\mathbf{Q}$ relevant to $\mathbf{T}$. Lastly, there is no causal link from $\mathbf{Q}$ to $\mathbf{R}$, as the effect of the quality of the published papers on the rank of the institution would be slow given the infrequent change in ranks.

### 4.3.2 Effect estimation

Recall that Q3 considers the papers that were posted online before the review process, which implies $\mathbf{P} = 1$. Since $\mathbf{P}$ is fixed, based on Figure 2, this implies there is no interference from $\mathbf{P}$ and $\mathbf{Q}$ on the causal estimand

| | | EC 2021 | ICML 2021 |
|---|---|---|---|
| 1 | # RESPONSES OVERALL | 449 | 7594 |
| 2 | # RESPONSES IN TIME BIN 1, 2, 3 | $159, 233, 57$ | $3799, 3228, 567$ |
| 3 | # RESPONSES WITH $r = 0$, 1 | $269, 180$ | $4936, 2658$ |
| 4 | ATE OF RANK ON VISIBILITY, $\tau$ IN EQUATION 2F | 0.036 | 0.024 |
| 5 | 95% CONFIDENCE INTERVAL ON ATE $\tau$ | [-0.038, 0.108] | [0.011, 0.036] |

Table 5: Outcome of analysis for research question Q3. Recall that for ICML, we consider the set of responses obtained for submissions that were available as preprints on arXiv. There were 1934 such submissions. The 95% confidence interval is obtained via bootstrapping.

in equation 1. In addition, we assume that there is no unmeasured confounding, that is, $\mathbf{V}^0, \mathbf{V}^1 \perp\!\!\!\perp \mathbf{R}|\mathbf{T}$. Consequently, to estimate $\tau$ we account for the role of time in the ATE.

To account for effect of rank on paper visibility mediated by time of posting, we gathered data on the time of posting for all preprints considered. There is ample variation in the time of posting papers online within the papers submitted to ICML and EC: some papers were posted right before the review process began while some papers were posted two years prior. To factor in the role of time, first we divide the responses into bins based on the number of days between the paper being posted online and the deadline for submitting responses to the visibility survey. Since similar conference deadlines arrive every three months roughly and the same conference recurs every one year, we binned the responses accordingly into three bins. Specifically, if the number of days between the paper being posted online and the survey response is less than 90, it is assigned to the first bin, we denote this range as $T_1$. Next if the number of days is between 90 and 365, the response is assigned to the second bin, range denoted by $T_2$, and otherwise, the response is assigned to the third bin, range denoted by $T_3$. Following this binning, we assume that for papers already posted online belonging to some rank group, time of posting does not affect the visibility of papers within the same bin: formally, for each $r \in \{0, 1\}$, $i \in \{1, 2, 3\}$, and $t_1, t_2 \in T_i$, we assume $\mathbb{E}[\mathbf{V}|\mathbf{T} = t_1, \mathbf{R} = r] = \mathbb{E}[\mathbf{V}|\mathbf{T} = t_2, \mathbf{R} = r]$.

Consequently, we derive the average treatment effect as:

$$\tau = \mathbb{E}\left[\mathbf{V}^1\right] - \mathbb{E}\left[\mathbf{V}^0\right] \tag{2a}$$

$$= \mathbb{E}_{\mathbf{T}}\left[\mathbb{E}\left[\mathbf{V}^1|\mathbf{T}\right]\right] - \mathbb{E}_{\mathbf{T}}\left[\mathbb{E}\left[\mathbf{V}^0|\mathbf{T}\right]\right] \tag{2b}$$

$$= \mathbb{E}_{\mathbf{T}}\left[\mathbb{E}\left[\mathbf{V}^1|\mathbf{T}, \mathbf{R} = 1\right]\right] - \mathbb{E}_{\mathbf{T}}\left[\mathbb{E}\left[\mathbf{V}^0|\mathbf{T}, \mathbf{R} = 0\right]\right] \tag{2c}$$

$$= \mathbb{E}\left[\mathbb{E}\left[\mathbf{V}|\mathbf{T}, \mathbf{R} = 1\right]\right] - \mathbb{E}\left[\mathbb{E}\left[\mathbf{V}|\mathbf{T}, \mathbf{R} = 0\right]\right] \tag{2d}$$

$$= \sum_{t \in T_1 \cup T_2 \cup T_3} \left(\mathbb{E}[\mathbf{V}|\mathbf{R} = 1, \mathbf{T} = t]\mathbb{P}(\mathbf{T} = t|\mathbf{R} = 1) - \mathbb{E}[\mathbf{V}|\mathbf{R} = 0, \mathbf{T} = t]\mathbb{P}(\mathbf{T} = t|\mathbf{R} = 0)\right) \tag{2e}$$

$$= \sum_{i \in \{1,2,3\}} \left(\mathbb{E}[\mathbf{V}|\mathbf{R} = 1, \mathbf{T} \in T_i]\mathbb{P}(\mathbf{T} \in T_i|\mathbf{R} = 1) - \mathbb{E}[\mathbf{V}|\mathbf{R} = 0, \mathbf{T} \in T_i]\mathbb{P}(\mathbf{T} \in T_i|\mathbf{R} = 0)\right). \tag{2f}$$

In the above derivation, we go from equation 2a to equation 2b using the law of iterated expectations, from equation 2b to equation 2c using the assumption of no unmeasured confounding ($\mathbf{V}^r \perp\!\!\!\perp \mathbf{R}|\mathbf{T}$) for $r \in \{0, 1\}$. Next, to derive equation 2d we apply the consistency assumption stating that for $r \in \{0, 1\}$, we have $\mathbf{V}^r = \mathbf{V}$ if $\mathbf{R} = r$. In equation 2e, the time variable is split up based on the binning approach described above. Finally, equation 2f comes from our assumption of time not affecting visibility for papers within the same bin. Note that this derivation also assumes that $\mathbb{P}(\mathbf{R} = r|\mathbf{T} \in T_i) > 0$ for all $r \in \{0, 1\}, i \in \{1, 2, 3\}$ for the conditional expectations in the derivation to be well-defined. If the 95% confidence interval of the average treatment effect $\tau$ does not contain zero, it implies that there exists a causal effect of papers' rank on its visibility. We obtain the confidence interval using bootstrapping.

### 4.3.3 Results of Q3

Table 5 depicts the results of the survey for research question Q3. We received 7594 responses and 449 responses for the visibility survey in ICML and EC respectively (Row 1). Recall that Q3 investigates the

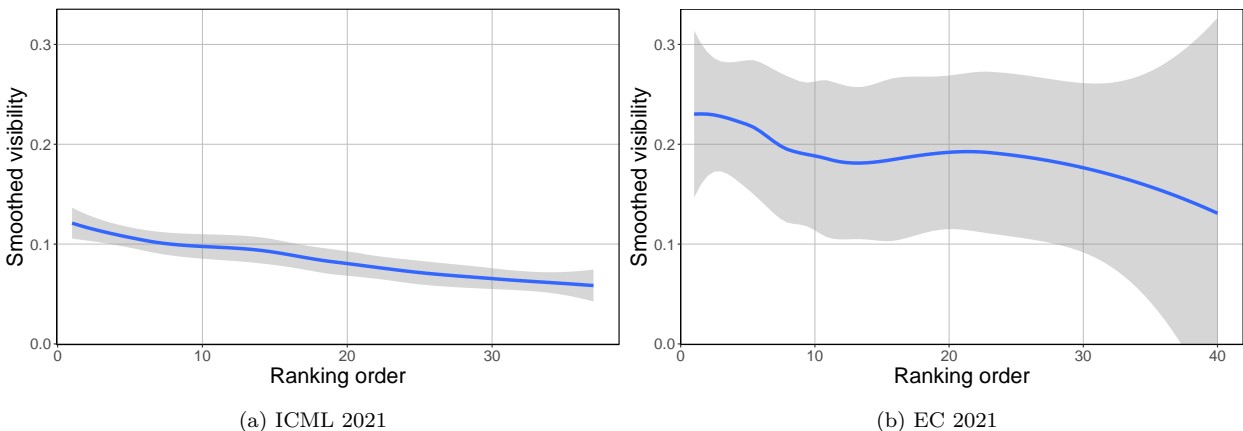

(a) ICML 2021

(b) EC 2021

Figure 3: Using responses obtained in visibility survey, we plot the papers' visibility against papers' associated rank with smoothing. On the x-axis, we have the ranking order as described in Figure 1. On the y-axis, smoothed visibility lies in $[0, 1]$. We use locally estimated smoothing to get the smoothed indicator for posting online across ranks, shown by the solid line, and a 95% confidence interval, shown by the grey region.

causal effect of papers' associated rank on their visibility, based on the causal model in Figure 2. As shown in Table 5 (Row 4 and Row 5), for papers submitted to the respective conference and posted online before the review process, we find a very weak positive effect of the papers' associated rank on their visibility. The 95% confidence interval around the average treatment effect for ICML does not contain zero, implying its statistical significance. Meanwhile, for EC, the ATE is not statistically significant. This may be attributed to the small sample size available in EC. Next, to understand the change in visibility enjoyed by preprints based on their associated rank, without rank binarization, we provide a visualization in Figure 3

Further, since the survey was optional in EC 2021, we analyse the difference between the responders and non-responders. Specifically, we looked at the distribution of the seniority of the reviewers that did and did not respond to the survey. We measure the seniority of the reviewers based on their reviewing roles, namely, (i) Junior PC member, (ii) Senior PC member, (iii) Area Chair, valued according to increasing seniority. Based on this measurement system, we investigated the distribution of seniority across the groups of responders and non-responders. We find that there is no significant difference between the two groups, with mean seniority given by 1.64 in the non-responders' group and 1.61 in the responders' group. The difference in the mean (0.03) is much smaller than the standard deviation in each group, which is 0.64 and 0.62 respectively.

## 5 Conclusions and limitations

To improve peer review and scientific publishing in a principled manner, it is important to understand the quantitative effects of the policies in place, and design policies in turn based on these quantitative measurements. In this work, we design experiments to understand the consequences of posting preprints online and related policies in the peer-review process. Specifically, this work addresses three questions on (Q1) reviewer practices around searching for preprints online, (Q2) current trends in preprint posting and their interaction with related peer-review policies, and (Q3) effect of institutional ranking on preprint visibility online. We now summarize our findings for each of these questions in order and discuss limitations therein.

Towards Q1, the data collected shows that more than a third of survey respondents self-report deliberately searching for their assigned papers online, thereby weakening the effectiveness of author anonymization in double-blind peer review. This finding has important implications for authors who perceive they may be at a disadvantage in the review process if their identity is revealed, in terms of their decision to post preprints online.

We discuss some caveats associated with this result. The survey corresponding to this finding was administered anonymously with a response rate of roughly 16% in ICML 2021 and 50% in EC 2021, which brings the possibility of selection bias in the results. However, it is relevant to note that our observed response rates are reasonably similar to the response rates commonly observed in surveys in peer review within the field of Computer Science. For example, an anonymous survey by Rastogi et al. (2024) in the UAI 2022 conference received a response rate of roughly 20%. Another survey of reviewers by Nobarany et al. (2016) in the 2011 CHI conference, which was opt-in and non-anonymous, had a response rate of 16%. Next, it is possible that the observed value of fraction of reviewers that searched for their assigned paper online in Table 1 might be an underestimate due to the following reasons: (i) Reviewers who did search for their assigned papers online may be more reluctant to respond to our survey for Q1. (ii) As we saw in Section 4.2.1, roughly 8% of reviewers in ICML 2021 had already seen their assigned paper before the review process began (Table 2 row 3). These reviewers' response to the Q1 survey would be "no" irrespective of what would have happened had they not seen their assigned paper before the review process.

Under Q2, we study the implications of the various embargo-style policies adopted by double-blind venues. These policies largely aim to prohibit authors from posting or advertising their work online for certain time periods around the peer-review process. The visibility survey in this study provides statistics about the number of preprints posted during and outside the embargo period, the visibility enjoyed by these papers, and the way it is achieved. These measurements can inform subsequent policy decisions. Furthermore, combining the result that more than a third of reviewers search online for their assigned paper, and the finding that a majority of the papers posted online were posted before the anonymity period suggests that conference policies designed towards banning authors from publicising their work on social media or from posting preprints online in a specific period of time before the review process may not be effective in maintaining double anonymity, since reviewers may still find these papers online if available. The Q2 survey saw a response rate of 100% in ICML 2021 but 55% in EC 2021 giving the possibility of response bias in EC.

Based on the analysis of Q3, we find evidence to support a weak effect of authors' affiliations' ranking on the visibility of their papers posted online before the review process. These papers represent the current preprint-posting habits of authors. Thus, authors from lower-ranked institutions get slightly less viewership for their preprints posted online compared to their counterparts from top-ranked institutions. For Q3, the effect size is significant in ICML (zero lies outside the 95% confidence interval on the ATE), but not in EC. A possible explanation for the difference is in the method of assigning rankings to institutions, described in Section 3.2. For ICML, the rankings used are directly related to past representation of the institutions at ICML (Ivanov, 2020). In EC, we used popular rankings of institutions such as QS rankings and CS rankings. In this regard, we observe that there is no clear single objective measure for ranking institutions in a research area. This leads to many ranking lists that may not agree with each other. Our analysis also suffers from this limitation. Another possible explanation for the difference is the small sample size in EC.

Next, while we try to carefully account for mediation by time of posting in our analysis for Q3, our study remains dependent on observational data. Thus, the usual caveat of unaccounted for confounding factors applies to our work. For instance, the topic of research may be a confounding factor in the effect of papers' rank on visibility: If authors from better-ranked affiliations work more on popular topics compared to others, then their papers would be read more widely. This could potentially increase the observed effect.

Finally, while our work finds dilution of anonymization in double-blind reviewing, any prohibition on posting preprints online comes with its own downsides. For instance, consider fields such as Economics where journal publication is the norm, which can often imply several years of lag between paper submission and publication. Double-blind venues must grapple with the associated trade-offs, and we conclude with a couple of suggestions for a better trade-off. First, many conferences, including but not limited to EC 2021 and ICML 2021, do not have clearly stated policies for reviewers regarding searching for papers online, and can clearly state as well as communicate these policies to the reviewers. Second, venues may consider policies requiring authors to use a different title and reword the abstract during the review process as compared to the versions available online, which may reduce the chances of reviewers discovering the paper or at least introduce some ambiguity if a reviewer discovers (a different version of) the paper online. Third, websites such as `openreview.net` support and facilitate posting anonymous preprints online while the articles undergo the OpenReview review

process. This allows authors to publicise the work anonymously and eventually associate the work with their name once the preprint is deanonymised.

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
