# OpenReview forum: "To ArXiv or not to ArXiv: A Study Quantifying Pros and Cons of Posting Preprints Online"
_TMLR — Rejected by TMLR_

### Review · Reviewer_Wu9r · 2025-05-30

**Summary Of Contributions:**

The paper uses a large-scale survey experiment on conference reviewing to study the interplay between preprint posting, author prestige, and reviewing. The primary dataset and the paper also complements it with auxiliary data on visibility and other things. The paper takes a formal causal inference approach in defining an estimand, model, and estimation strategy.

**Audience:**

Yes

**Claims And Evidence:**

No

**Requested Changes:**

Quality affects visibility. I would be eager to read a revised version of the paper that adopts any strategy other than assuming this away, and I would review such a paper openly and generously. Unfortunately though, I think this assumption is just too unbelievable (as the paper itself says).

**Strengths And Weaknesses:**

Strengths: The paper is addressing an interesting and important topic in a data-intensive and formal way. I want to specifically commend the authors on their data efforts. I know how difficult and often unappreciated data engineering can be and I can tell that the authors have done a lot of work behind the scenes to collect high quality and relevant data to study this problem. The paper is also strong in how it clearly states its research questions and operationalizes its concepts clearly, formalizing them as variables in a causal model and setting up the question as a target estimand.

Weaknesses: There remains one glaring weakness, which is the main assumption upon which causal identification rests---i.e., that paper quality does not causally affect visibility except through the decision to post online. I hate to be a curmudgeon about this, because often we cannot do better in practice than make and clearly articulate slightly unrealistic assumptions and proceed. However, the main assumption in the paper is so unrealistic that the paper itself contradicts it. Take the following sentence in the paper:

"[...] a majority of the reviewers learnt about the preprints they were queried about from first hand sources such as arXiv and talk announcements. This finding suggests that on most occasions people view a paper (and its authors) before knowing the paper’s quality."

While this sentence is supposed to support the assumption, it gives as an example that reviewers learned about preprints via talk announcements. Clearly, the quality of a preprint affects whether the authors of the paper are invited to give talks about the preprint, and therefore quality also affects whether reviewers see it (via those talks).

The authors need to do substantial work to tighten this up. Obviously it is not acceptable for the paper itself to provide counterexamples to the main assumption. But simply editing this particular sentence will not be sufficient. The authors need to find a way to analyze the data without assuming away the arrow between quality and visibility; this arrow is just too obviously present.

---

> ### Author Response · Authors · 2025-07-14
> **Author response to Reviewer Wu9r**
>
> Thank you for this suggestion. In response to your feedback, we conducted sensitivity analyses to understand how much unobserved confounding by ‘paper quality’ can affect the results. That is, we conducted new experiments to understand the potential role of the edge between Quality(Q) and Visibility(V), that we had previously presumed to be absent, in the ATE estimation. In the final manuscript, we will reflect this change as follows:
>
> **Changes in manuscript proposed for the camera ready version**:
> In the camera ready version, we will provide the sensitivity analysis detailed below to show under what level of strength between R→Q and Q→V does the result in Table 5 change. The sensitivity analysis and its results are described next.
>
> **Description of sensitivity analysis conducted**:
> Briefly, to understand the role of effect of Quality(Q) on Visibility(V) on the effect of Rank(R) on Visibility(V), we simulate Q (as it is unmeasurable in our setting) under several modeling assumptions, and then incorporate it in the ATE estimation to see how the ATE varies under the different modeling assumptions for Q.
>
> * **Step 1**: Modeling and simulating Q: We model Q as a binary variable, based on its relationship with R and V. For this, we introduce variables x,y,z each in range [0.0,0.5]. Each value combination of x,y,z represents how strongly Q is influenced by R and influences V, as shown in the following modeling assumptions:
>      - (i) P (Q=1 | R=0, V=0) = 0.5,
>     - (ii) P (Q=1 | R=0, V=1) = 0.5+x,
>     - (iii) P(Q=1 | R=1, V=0) = 0.5+y,
>     - (iv) P(Q=1 | R=1, V=1) = 0.5 + z.
>
> * **Step 2**: Identifying the modeling assumptions that yield result reversal or invalidation. Specifically, for ICML we identify the value of {x,y,z} under which the ATE is not statistically significant. Next for EC, we identify the value of {x,y,z} for which the ATE is statistically significantly in the opposite direction (compared to the ATE for EC that does not consider interference by Q)
>
>
> **Sensitivity analysis results**:
> For ICML, we note here some values of {x,y,z} for which the corresponding ATE is such that its 95% confidence interval contains zero (ie is statistically insignificant)
>
> * (x,y,z) = (0.05, 0.30, 0.45). ATE= 0.01240, 95% confidence interval: [-0.0006, 0.0252]
> * (x,y,z) = (0.10, 0.25, 0.45). ATE= 0.01072, 95% confidence interval: [-0.0024, 0.0237]
> * (x,y,z) = (0.10, 0.30, 0.40). ATE= 0.01285, 95% confidence interval: [-0.0009, 0.0257]
> * (x,y,z) = (0.20, 0.30, 0.35). ATE= 0.01167,  95% confidence interval: [-0.0024, 0.0254]
>
> For EC, we see that there is no value of $x,y,z \in[0,0.5]^3$ for which ATE is statistically significantly negative. Next, we see that for the following values the ATE becomes statistically significantly positive.
>
> * (x,y,z) =  (0.00, 0.40, 0.00). ATE= 0.09728, 95% confidence interval: [0.0178, 0.1723]
> * (x,y,z) =  (0.00, 0.40, 0.10). ATE= 0.09073, 95% confidence interval: [0.0074, 0.1700]
> * (x,y,z) = (0.00, 0.40, 0.20). ATE= 0.08503, 95% confidence interval: [0.0010, 0.1668]
> * (x,y,z) =  (0.10, 0.40, 0.00). ATE= 0.08428, 95% confidence interval: [0.0034,  0.164]
>
> **Takeaway**: These values show that the relationship between Q and V needs to be quite strong for the result to change (i.e. become statistically insignificant) for ICML, this is indicated by the high values of  x and z which are indicators of the strength of effect of Q on V.  Similarly, for EC we see that for the result changes for only high values of y (P(Q=1 | R=1, V=0) = 0.9)  which are unlikely to occur in practice. Thus, with this sensitivity analysis, we conclude that our results are robust to confounding by Quality. This result is further supported by our finding in EC via survey that 55% of the preprints seen prior to the review period were made visible via arXiv.
>
> Lastly, we note that this additional analysis will be added to the camera ready version.

---

### Review · Reviewer_DMe4 · 2025-06-01

**Summary Of Contributions:**

The submission has three major contributions. Firstly, an anonymous survey of ICML 2021 and EC 2021 reviewers reveals that over one‐third of them admit they have searched for their assigned papers online, challenging the efficacy of double‐blind review. Secondly, the authors scraped arXiv and manually checked EC submissions, revealing that about 37 % of papers were live on arXiv before review. Moreover, a visibility survey shows that “one‐month preprint bans” do little to prevent reviewer access. Lastly, the authors conducted a causal analysis comparing high‐ranking and low‐ranking institutions and found only a very small early‐visibility advantage for top institutions (ATE about 0.024 at ICML, 0.036 at EC), much smaller than anecdotal claims.

**Audience:**

Yes

**Broader Impact Concerns:**

There is no significant negative impact noted.

**Claims And Evidence:**

Yes

**Requested Changes:**

1. Check causal assumptions by running a data‐driven method (e.g., PC algorithm) or a sensitivity analysis to ensure no hidden path from paper quality to early visibility.
2. Replace the binary “top 50 or not” split with finer rank brackets or a continuous rank model (for example, spline regression) to capture nonlinear prestige effects.

Overall, I recommend acceptance according to the reviewers’ guidelines.

**Strengths And Weaknesses:**

Strengths:
1. The study conducts large surveys at top conferences to track reviewer behavior and preprint visibility. More than one third of surveyed reviewers admit they searched for assigned papers online, providing concrete evidence (rather than inference) that double‐blind review can be violated.
2. By scraping arXiv and checking EC submissions, the paper finds that about 37% of papers were publicly available before review. A follow‐up visibility survey shows that “one‐month preprint bans” do little to stop reviewers from seeing preprints.
3. A simple causal analysis contrasting high‐ranking versus low‐ranking institutions finds only a small early‐visibility advantage for top institutions. This contradicts larger anecdotal claims about prestige bias.
4. The paper offers practical suggestions: tell reviewers explicitly not to search online, use different titles or abstracts during review, or host preprints on anonymous platforms. These ideas address how to support preprinting while preserving anonymity.

Weaknesses:
1. Survey response rates were low (16 percent for ICML and 51 percent for EC), which could bias estimates of how often reviewers search for papers.
2. None of the proposed policy recommendations (such as a “no‐search” mandate or alternate titles) are tested experimentally. Without a trial or follow‐up study, it is unclear whether these ideas actually reduce de‐anonymization.
3. The causal model assumes that paper quality does not directly affect early visibility, and that posting time fully mediates the effect of rank. These assumptions are not checked using data‐driven causal discovery (for example, the PC algorithm) or sensitivity analysis, so unmeasured factors like hot research topics or author networks could still bias results.
4. Treating institution rank as a simple “top 50 or not” split may hide more nuanced effects. A finer stratification (for example, three or more rank brackets) or a continuous analysis could reveal non‐linear prestige effects.

---

> ### Author Response · Authors · 2025-07-14
> **Author response to Reviewer DMe4**
>
> Thank you for your careful feedback! We have incorporated it and updated the manuscript -- we provide a point by point response here.
>
> **Weakness 1**: Survey response rates were low (16 percent for ICML and 51 percent for EC), which could bias estimates of how often reviewers search for papers.
>
> **Author response*: We discuss this in the manuscript in the discussion section, text reproduced here:
>
> “The survey corresponding to this finding was administered anonymously with a response rate of roughly 16% in ICML 2021 and 50% in EC 2021, which brings the possibility of selection bias in the results. However, it is relevant to note that our **observed response rates are reasonably similar to the response rates commonly observed in surveys in peer review within the field of Computer Science**. For example, an anonymous survey by Rastogi et al. (2024) in the **UAI 2022 conference received a response rate of roughly 20%**. Another survey of reviewers by Nobarany et al. (2016) **in the 2011 CHI conference, which was opt-in and non-anonymous, had a response rate of 16%**. Next, it is possible that the observed value of the fraction of reviewers that searched for their assigned paper online in Table 1 might be an underestimate due to the following reasons: (i) Reviewers who did search for their assigned papers online may be more reluctant to respond to our survey for Q1. (ii) As we saw in Section 4.2.1, roughly 8% of reviewers in ICML 2021 had already seen their assigned paper before the review process began (Table 2 row 3). These reviewers’ response to the Q1 survey would be “no” irrespective of what would have happened had they not seen their assigned paper before the review process.”
>
> **Weakness 2**: None of the proposed policy recommendations (such as a “no‐search” mandate or alternate titles) are tested experimentally. Without a trial or follow‐up study, it is unclear whether these ideas actually reduce de‐anonymization.
>
> **Author response**: This paper is focused on collecting data to draw evidence based policy recommendations. The paper does not make any claims regarding the efficacy of the policy suggestions. Future work would be well placed to answer the question of efficacy of the proposed policy updates.
>
> **Weakness 4 and Requested change 2**: Treating institution rank as a simple “top 50 or not” split may hide more nuanced effects. A finer stratification (for example, three or more rank brackets) or a continuous analysis could reveal non‐linear prestige effects.
>
> **Author response**: We designed the approach of binarizing the rank data drawing from prior work in this space (Tomkins et al 2017). You are correct to point out that binarization may hide some trends. For this, we computed and visualized the trend of paper visibility for all unique ranks available in our data in Figure 3.

---

> > ### Author Response · Authors · 2025-07-14
> > **Continuation of Response to Reviewer DMe4**
> >
> > **Requested Change 1 and Weakness 3**: Check causal assumptions by running a data‐driven method (e.g., PC algorithm) or a sensitivity analysis to ensure no hidden path from paper quality to early visibility.
> >
> >  **Author response** :
> >
> > Thank you for this suggestion. In response to your feedback, we conducted sensitivity analyses to understand how much unobserved confounding by ‘paper quality’ can affect the results. That is, we conducted new experiments to understand the potential role of the edge between Quality(Q) and Visibility(V), that we had previously presumed to be absent, in the ATE estimation. In the final manuscript, we will reflect this change as follows:
> >
> > **Changes in manuscript proposed for the camera ready version**:
> > In the camera ready version, we will provide the sensitivity analysis detailed below to show under what level of strength between R→Q and Q→V does the result in Table 5 change. The sensitivity analysis and its results are described next.
> >
> > **Description of sensitivity analysis conducted**:
> > Briefly, to understand the role of effect of Quality(Q) on Visibility(V) on the effect of Rank(R) on Visibility(V), we simulate Q (as it is unmeasurable in our setting) under several modeling assumptions, and then incorporate it in the ATE estimation to see how the ATE varies under the different modeling assumptions for Q.
> >
> > * **Step 1**: Modeling and simulating Q: We model Q as a binary variable, based on its relationship with R and V. For this, we introduce variables x,y,z each in range [0.0,0.5]. Each value combination of x,y,z represents how strongly Q is influenced by R and influences V, as shown in the following modeling assumptions:
> >      - (i) P (Q=1 | R=0, V=0) = 0.5,
> >     - (ii) P (Q=1 | R=0, V=1) = 0.5+x,
> >     - (iii) P(Q=1 | R=1, V=0) = 0.5+y,
> >     - (iv) P(Q=1 | R=1, V=1) = 0.5 + z.
> >
> > * **Step 2**: Identifying the modeling assumptions that yield result reversal or invalidation. Specifically, for ICML we identify the value of {x,y,z} under which the ATE is not statistically significant. Next for EC, we identify the value of {x,y,z} for which the ATE is statistically significantly in the opposite direction (compared to the ATE for EC that does not consider interference by Q)
> >
> >
> > **Sensitivity analysis results**:
> > For ICML, we note here some values of {x,y,z} for which the corresponding ATE is such that its 95% confidence interval contains zero (ie is statistically insignificant)
> >
> > * (x,y,z) = (0.05, 0.30, 0.45). ATE= 0.01240, 95% confidence interval: [-0.0006, 0.0252]
> > * (x,y,z) = (0.10, 0.25, 0.45). ATE= 0.01072, 95% confidence interval: [-0.0024, 0.0237]
> > * (x,y,z) = (0.10, 0.30, 0.40). ATE= 0.01285, 95% confidence interval: [-0.0009, 0.0257]
> > * (x,y,z) = (0.20, 0.30, 0.35). ATE= 0.01167,  95% confidence interval: [-0.0024, 0.0254]
> >
> > For EC, we see that there is no value of $x,y,z \in[0,0.5]^3$ for which ATE is statistically significantly negative. Next, we see that for the following values the ATE becomes statistically significantly positive.
> >
> > * (x,y,z) =  (0.00, 0.40, 0.00). ATE= 0.09728, 95% confidence interval: [0.0178, 0.1723]
> > * (x,y,z) =  (0.00, 0.40, 0.10). ATE= 0.09073, 95% confidence interval: [0.0074, 0.1700]
> > * (x,y,z) = (0.00, 0.40, 0.20). ATE= 0.08503, 95% confidence interval: [0.0010, 0.1668]
> > * (x,y,z) =  (0.10, 0.40, 0.00). ATE= 0.08428, 95% confidence interval: [0.0034,  0.164]
> >
> > **Takeaway**: These values show that the relationship between Q and V needs to be quite strong for the result to change (i.e. become statistically insignificant) for ICML, this is indicated by the high values of  x and z which are indicators of the strength of effect of Q on V.  Similarly, for EC we see that for the result changes for only high values of y (P(Q=1 | R=1, V=0) = 0.9)  which are unlikely to occur in practice. Thus, with this sensitivity analysis, we conclude that our results are robust to confounding by Quality. This result is further supported by our finding in EC via survey that 55% of the preprints seen prior to the review period were made visible via arXiv.
> >
> > Lastly, we note that this additional analysis will be added to the camera ready version.

---

### Review · Reviewer_Z4rH · 2025-06-11

**Summary Of Contributions:**

This paper presents a study to quantify some of the pros and cons of authors posting preprints on ArXiv.  The papers methodology relies on survey responses collected from 2 conferences and a causal analysis of the impact of paper rank (author rank?) on paper visibility.

**Audience:**

Yes

**Broader Impact Concerns:**

The potential broader impact of this work lies in better informed decision-making by conference chairs with regards to allowing or disallowing arxiv posting of pre-prints.  Risks include overreliance in decision-making on a single analysis.

**Claims And Evidence:**

No

**Requested Changes:**

Would be good to clarify in section 4.3 whether or when there is a distinction between "the paper's associated rank" and "rank of the authors' affiliations".  are they the same? is the paper's rank

The authors description of the model assumptions (the missing edges in Fig 2) are plausible, but not convincing.  I'd recommend running the analysis with multiple distinct assumptions as a sensitivity analysis. I would also recommend running additional sensitivity analyses under simulated unobserved confounding to show how the observed effect (the confidence interval in Table 5 seems to only account for statistical variance and not for incorrect causal assumptions).

I would appreciate if the authors interpreted their results to present clearer implications for conference policies towards ArXiv posting.

**Strengths And Weaknesses:**

Overall, the paper presents some interesting findings.  However, I have a number of methodological questions, especially with regards to the survey response rate for Q2 and the some of the data selection and modeling for Q3.  In addition, the implications of the findings on the motivating question (to allow or disallow arxiv postings) is unclear.


Strengths:
- the research question is interesting and I appreciate the effort and logistical challenges the authors must have overcome to execute the data gathering for this analysis.
- the paper includes a variety of methodologies, including causal analysis of the influence of authors rank on visibliity of papers.

Weaknesses:
- The survey question response rate of 100% for Q2 seems too good to be true. Is there an explanation for why there was a perfect response rate?
- The causal graph presented in Q3's analysis is not entirely convincing.  Furthermore, why is this phrasing of Q3 the right question? Shouldn't the primary concern be about the influence of arxiv posting on reviewer bias?
- The conclusions of the paper are unclear.  I.e., what are the implications of these findings for conference chairs?  Should they allow arxiv posting or not?


Questions:
 - why was the ICML survey response rate 100% for Q2 (Sec 4.2)?  Was it required somehow? 100% response rate seems too good to be true, especially when we see that Q1 received a response rate of only 16%

- There is a value P (Posted online) in Figure 2, but the 2nd sentence of 4.3.1 says "we consider all papers ... posted online before the review process".  If the only data being considered is for papers where P=1, this will bias the analysis results.

---

> ### Author Response · Authors · 2025-07-14
> **Response to Reviewer Z4rH**
>
> Thank you for your careful feedback! We have incorporated it  and updated the manuscript -- we provide a point by point response  here.
>
> **Question 1**: why was the ICML survey response rate 100% for Q2 (Sec 4.2)? Was it required somehow? 100% response rate seems too good to be true, especially when we see that Q1 received a response rate of only 16%
>
>
> **Author response**: The survey question for Q2 was required in ICML as it was a part of the reviewer response form. In the current manuscript this is mentioned in Section 3.2 under the paragraph heading ‘Paper’s visibility’.  We have further emphasised this in the updated manuscript in Section 4.2. We reproduce the new text here:
>
> “Recall that while the survey question for Q2 was a mandatory part of the ICML reviewer response form, it was asked as a part of an optional survey in EC.”
>
> Additionally , for Q1, we had to conduct a separate survey to elicit information about whether reviewers looked up their assigned papers on the Internet. In this survey, reviewers had the option to not respond and remain anonymous.
> Noteworthily, the observed response rate of 16% for the anonymous survey for Q1 is reasonably similar to the response rates commonly observed in surveys in peer review within the field of Computer Science. For example, a survey of reviewers by Nobarany et al. (2016) in the 2011 CHI conference, which was opt-in and non-anonymous, had a response rate of 16%. Another survey by Rastogi et al. (2024) in the UAI 2022 conference which was anonymous received a response rate of roughly 20%.
>
>
> **Question 2**: There is a value P (Posted online) in Figure 2, but the 2nd sentence of 4.3.1 says "we consider all papers ... posted online before the review process". If the only data being considered is for papers where P=1, this will bias the analysis results.
>
> **Author response**: Figure 2 is the full causal model considering all variables in our setting. For the causal effect question in Q3, we only consider papers with P=1. As P is fixed in the setting considered for Q3, we remove the edges connected to P for conducting the analysis for Q3.
>
> **Requested Changes 1**: Would be good to clarify in section 4.3 whether or when there is a distinction between "the paper's associated rank" and "rank of the authors' affiliations". are they the same? is the paper's rank
>
> **Author response**: Thank you for pointing that out! They are the same, we have clarified that in the paper in Section 3.2. We reproduce the text here:
> “The rank of an author's affiliation is a measure of author's prestige that, in turn, is transferred to the author's paper. In this term we refer to this rank as 'the rank associated with a paper' or ‘the rank of the authors' affiliations'. ”
>
> **Requested changes 2**: I would appreciate if the authors interpreted their results to present clearer implications for conference policies towards ArXiv posting.
>
> **Author response**: Implications for policy are discussed on Page 2 and 3. Text reproduced here, with policy recommendations highlighted:
>
> * [Q1] Searching for paper online
> * -- More than a third of survey respondents self-report searching for their assigned papers online.
> * -- **In double-blind review processes, reviewers should be explicitly instructed to not search for their assigned papers on the Internet.**
> * [Q2] Trends and visibility of preprints
> * -– On average, authors posting preprints online receive viewership from 8% of relevant researchers in ICML and 20% in EC before the conclusion of the review process.
> * -– Certain conferences ban posting preprints a month before the submission deadline. In ICML and EC (which did not have such a ban), more than 50% of preprints posted online where posted before the one month period, and these enjoyed a visibility of 8.6% and 18% respectively.
> * -– **Conference policies designed towards banning authors from publicising their work on social media or posting preprints before the review process may have only limited effectiveness in maintaining anonymity.**

---

> ### Author Response · Authors · 2025-07-14
> **Continuation of response to Reviewer Z4rH**
>
> **requested changes 3** : The authors description of the model assumptions (the missing edges in Fig 2) are plausible, but not convincing. I'd recommend running the analysis with multiple distinct assumptions as a sensitivity analysis. I would also recommend running additional sensitivity analyses under simulated unobserved confounding to show how the observed effect (the confidence interval in Table 5 seems to only account for statistical variance and not for incorrect causal assumptions).
>
> **Author response** :
>
> Thank you for this suggestion. In response to your feedback, we conducted sensitivity analyses to understand how much unobserved confounding by ‘paper quality’ can affect the results. That is, we conducted new experiments to understand the potential role of the edge between Quality(Q) and Visibility(V), that we had previously presumed to be absent, in the ATE estimation. In the final manuscript, we will reflect this change as follows:
>
> **Changes in manuscript proposed for the camera ready version**:
> In the camera ready version, we will provide the sensitivity analysis detailed below to show under what level of strength between R→Q and Q→V does the result in Table 5 change. The sensitivity analysis and its results are described next.
>
> **Description of sensitivity analysis conducted**:
> Briefly, to understand the role of effect of Quality(Q) on Visibility(V) on the effect of Rank(R) on Visibility(V), we simulate Q (as it is unmeasurable in our setting) under several modeling assumptions, and then incorporate it in the ATE estimation to see how the ATE varies under the different modeling assumptions for Q.
>
> * **Step 1**: Modeling and simulating Q: We model Q as a binary variable, based on its relationship with R and V. For this, we introduce variables x,y,z each in range [0.0,0.5]. Each value combination of x,y,z represents how strongly Q is influenced by R and influences V, as shown in the following modeling assumptions:
>      - (i) P (Q=1 | R=0, V=0) = 0.5,
>     - (ii) P (Q=1 | R=0, V=1) = 0.5+x,
>     - (iii) P(Q=1 | R=1, V=0) = 0.5+y,
>     - (iv) P(Q=1 | R=1, V=1) = 0.5 + z.
>
> * **Step 2**: Identifying the modeling assumptions that yield result reversal or invalidation. Specifically, for ICML we identify the value of {x,y,z} under which the ATE is not statistically significant. Next for EC, we identify the value of {x,y,z} for which the ATE is statistically significantly in the opposite direction (compared to the ATE for EC that does not consider interference by Q)
>
>
> **Sensitivity analysis results**:
> For ICML, we note here some values of {x,y,z} for which the corresponding ATE is such that its 95% confidence interval contains zero (ie is statistically insignificant)
>
> * (x,y,z) = (0.05, 0.30, 0.45). ATE= 0.01240, 95% confidence interval: [-0.0006, 0.0252]
> * (x,y,z) = (0.10, 0.25, 0.45). ATE= 0.01072, 95% confidence interval: [-0.0024, 0.0237]
> * (x,y,z) = (0.10, 0.30, 0.40). ATE= 0.01285, 95% confidence interval: [-0.0009, 0.0257]
> * (x,y,z) = (0.20, 0.30, 0.35). ATE= 0.01167,  95% confidence interval: [-0.0024, 0.0254]
>
> For EC, we see that there is no value of $x,y,z \in[0,0.5]^3$ for which ATE is statistically significantly negative. Next, we see that for the following values the ATE becomes statistically significantly positive.
>
> * (x,y,z) =  (0.00, 0.40, 0.00). ATE= 0.09728, 95% confidence interval: [0.0178, 0.1723]
> * (x,y,z) =  (0.00, 0.40, 0.10). ATE= 0.09073, 95% confidence interval: [0.0074, 0.1700]
> * (x,y,z) = (0.00, 0.40, 0.20). ATE= 0.08503, 95% confidence interval: [0.0010, 0.1668]
> * (x,y,z) =  (0.10, 0.40, 0.00). ATE= 0.08428, 95% confidence interval: [0.0034,  0.164]
>
> **Takeaway**: These values show that the relationship between Q and V needs to be quite strong for the result to change (i.e. become statistically insignificant) for ICML, this is indicated by the high values of  x and z which are indicators of the strength of effect of Q on V.  Similarly, for EC we see that for the result changes for only high values of y (P(Q=1 | R=1, V=0) = 0.9)  which are unlikely to occur in practice. Thus, with this sensitivity analysis, we conclude that our results are robust to confounding by Quality. This result is further supported by our finding in EC via survey that 55% of the preprints seen prior to the review period were made visible via arXiv.
>
> Lastly, we note that this additional analysis will be added to the camera ready version.

---

### Decision · Action_Editor_EmoX · 2025-11-05

**Recommendation:** Reject

**Additional Comments:**

Overall, in their final assessments, two of the three reviewers suggested that the paper was not yet ready for publication. The third reviewer suggested acceptance but noted similar limitations as the other two.

I do find there's a path toward publication, but it requires another round of reviewing. I hesitated between "reject" and "minor revision," but I do find it essential to have the next version of the paper re-read by (ideally the same) two experts.

While this might not be helpful to the authors, I apologize for not coming to this final decision more quickly.


---

Below, I also suggest other relatively minor changes:

+ 'One reviewer notes that: the "institutional rank on preprint visibility" is not the same as the conclusion that "the benefits of posting preprints online are nearly comparable for both groups.". That conclusion would require estimating the counterfactual of paper visibility for both groups if preprints were not posted (only published at a conference).'
- I suggest adding relevant references in this paragraph: "Independently, authors who perceive they may be at a disadvantage in the review process if their identity is revealed face a dilemma regarding posting their work online. On one hand, if they post preprints online, they are at risk of de-anonymization in the review process, if their reviewer searches for their paper online. Past research suggests that if such de-anonymization happens, reviewers may get biased by the author identity, with the bias being especially harmful for authors from less
prestigious organizations. On the other hand, if they choose to not post their papers online before the review process, they stand to lose out on
viewership and publicity for their papers."
- "online where posted" -> "online were posted"
- "On the other hand, for ICML, owing to its large size, we checked whether a submitted paper was available on arXiv (arxiv.org). ArXiv is
  the predominant platform for pre-prints in machine learning; hence we used availability on arXiv as a proxy indicator of a paper’s availability on the Internet." -> slightly different titles
- "Rank associated with a paper" -> why different procedures for ICML and EC? It would be worth providing a brief justification for ICML's
  procedure (and/or comparing it to the QS rankings).
- Table 4: Archiv -> ArXiv

**Audience:**

Yes

**Audience Explanation:**

This work is of clear interest to all researchers in the TMLR audience, and it provides actionable insights to conference (and journal) organizers, including the conference boards.

**Claims And Evidence:**

No

**Claims Explanation:**

In their final assessments, the reviewers outlined a few remaining concerns:

1) Reliability of the survey response
2) The causal graph and third question (Q3).

For 1, the reviewers mention the two different methodologies (one for EC and one for ICML) and question the reliability of the results. I don't disagree, and while it's hard to be definitive either way and the response rate is reasonably low, the authors at least are clear about the limitations. They also discuss why the effect may even be underestimated. In any case, I find that, regardless of the true percentage of "de-anonymized papers", the results presented in this paper are convincing from a policy standpoint. For example, with those results, PCs could update their policies and/or be compelled to conduct additional surveys to support and augment the observations made in the current paper.

For the latter (2), all three reviewers discuss the assumptions underlying the proposed causal graph (Figure 2) and, in particular: a) how institution rank is used and; b) the missing edge between quality and visibility.

Regarding the institution rank, there are several ways the authors could have proceeded (e.g., with or without binarization). I think further justifying the different methods used for EC vs. ICML, along with their possible downstream impact, would help, but that's fairly minor.

The main remaining issue concerns the assumption that paper quality does not causally affect visibility. The reviewers recognize the value of the new sensitivity analysis, but also believe it should be expanded. I'm pasting directly from a reviewer comment: "[I suggest] a more thorough sensitivity analysis that includes settings where baseline quality is assumed to be rare (as is likely the case). In the reported sensitivity analysis, baseline quality is assumed to be a coin-flip, which seems high given that we are talking about arXiv. Second, I would want to see how the paper's prose is revised to not center this assumption as much." I also find that the current version of the paper could better justify this decision (missing link). For example, while unmeasured, Tables 3 & 4 don't rule out the possibility that quality plays a role (e.g., higher-quality work might be more likely to be presented at talks or shared within labs). This was also highlighted in the initial reviews.

**Resubmission Of Major Revision:**

The authors may consider submitting a major revision at a later time.